# Large-Scale Field Trials of an *Eimeria* Vaccine Induce Positive Effects on the Production Index of Broilers

**DOI:** 10.3390/vaccines12070800

**Published:** 2024-07-19

**Authors:** Binh T. Nguyen, Dongjean Yim, Rochelle A. Flores, Seung Yun Lee, Woo H. Kim, Seung-Hwan Jung, Sangkyu Kim, Wongi Min

**Affiliations:** 1College of Veterinary Medicine & Institute of Animal Medicine, Gyeongsang National University, Jinju 52828, Republic of Korea; thanhbinhcnty@gmail.com (B.T.N.); floresrochellea@gmail.com (R.A.F.); seungyun0218@gnu.ac.kr (S.Y.L.); woohyun.kim@gnu.ac.kr (W.H.K.); 2Hoxbio, Business Center, Gyeongsang National University, Jinju 52828, Republic of Korea; dongjean1@naver.com; 3MSD Animal Health Korea Ltd., Seoul Square, Jung-gu, Seoul 04637, Republic of Korea; seung.hwan.jung@msd.com (S.-H.J.); sangkyu.kim@msd.com (S.K.)

**Keywords:** chickens, coccidiosis vaccine, broiler farms, evaluation of parameters, fecal microbiota

## Abstract

Live coccidiosis vaccines have mainly been used to reduce *Eimeria* species infection, which is considered the most economically important disease in the poultry industry. Evaluation data on vaccine effectiveness through large-scale field experiments are lacking, especially in broilers. Thus, the effectiveness of a commercial coccidiosis vaccine was evaluated in approximately 900,000 chicks reared on three open-broiler farms where coccidiosis is prevalent. The vaccine’s effectiveness after vaccination of 1-day-old chicks was monitored using three parameters (lesion score, fecal oocyst shedding, and production index, PI) in nine trials performed three times on each farm. Lesion scores were confirmed in three different areas of the intestine because the vaccine contained four *Eimeria* species. The average lesion scores were 0.36 to 0.64 in the duodenal region, 0.30 to 0.39 in the jejuno-ileal region, and 0.18 to 0.39 in the cecal region. The average fecal oocyst shedding rate ranged from 19,766 to 100,100 oocysts per gram, showing large variations depending on farms and buildings within the farm. Compared with the PI of the previous 9–10 trials on each farm, the PI increased by 2.45 to 23.55. Because of the potential for perturbation of the fecal microbiota by live coccidiosis vaccines, the fecal microbiota was investigated using 16S rRNA microbial profiling. Although the β-diversity was significantly different in distribution and relative abundance among farms (PERMANOVA, pseudo-F = 4.863, *p* = 0.009), a Kyoto Encyclopedia of Genes and Genomes pathway analysis found no significant bacterial invasion of the epithelial cell pathway across farms. This large-scale field trial of a live *Eimeria* vaccine indicates that coccidiosis vaccines can have meaningful effects on the poultry industry and could be used as an alternative to the prophylactic use of anticoccidial drugs under field conditions.

## 1. Introduction

Coccidiosis is a major enteric disease affecting the productivity, health, and welfare of chickens and is considered the most economically important disease in the poultry industry. The estimated global cost of coccidiosis control, based on 2016 figures, was GBP 10.36 billion [1], and the financial cost of coccidiosis per chicken raised in 2022 was GBP 0.30 [2]. The disease is caused by any one or a combination of several species of the apicomplexan protozoa *Eimeria,* which are known to invade specific regions of the gastrointestinal tract in a host-specific manner. Fecal samples from farms typically contain an average of three to four *Eimeria* species [3,4]. Moreover, the infection rates of this disease can reach up to 90% in industrial farms worldwide [5,6]. The most frequently occurring species in farms are *E. acervulina*, *E. maxima,* and *E. tenella*, although there may be differences depending on the production type and geographic location [6,7,8]. Each species of *Eimeria* invades the host’s intestinal epithelial tissue, eliciting a variety of clinical signs in infected chickens depending on the infectious dose of sporulated oocysts, including the destruction of the villi and epithelial cells, the induction of inflammatory responses, and the suppression of bone development, ultimately leading to necrotic intestinal lesions, reduced feed conversion and body weight gain, and increased mortality and susceptibility to secondary pathogens. Consequently, *Eimeria* infection results in severe negative effects on the production index (PI) [9,10,11,12,13].

Increasing protein requirements along with the growth of the human population are leading to increases in intensive farming systems and higher stocking densities on farms, which are expected to increase the potential for economic losses due to *Eimeria* infection. This economically costly disease is controlled mainly by the prophylactic use of anticoccidial drugs in feed or water [3]. However, the application of anticoccidial drugs has led to the appearance of drug-resistant strains over time in different parts of the world and has reduced the efficacy of anticoccidials [8,14,15]. Therefore, to address the growing consumer demand for antibiotic-free poultry products and the appearance of drug-resistant strains, much effort has been expended in developing alternative control strategies to antibiotics, including dietary supplements such as phytochemicals, probiotics, and prebiotics, good husbandry systems, and vaccines [3,6,12,16,17,18]. Of these, vaccines that do not cause any resistance are receiving great attention in the chicken industry to prevent coccidiosis.

Although subunit or recombinant vaccines may be highly desirable to protect against multiple *Eimeria* species, they still have many drawbacks, including partial protective efficacy, the vaccine administration route, and high cost [19,20,21,22]. Therefore, live or attenuated vaccines have been used to control coccidiosis in commercial poultry industries since the 1950s and have seen tremendous success in disease prevention [23,24]. However, *Eimeria* species induce species-specific mucosal immunity with little or no cross protection against heterologous species. Live vaccines require the production of *Eimeria* oocysts in chickens, and because farm droppings contain an average of three to four *Eimeria* species, vaccines typically contain more than three *Eimeria* species, requiring considerable time, labor, and cost [4,8]. To induce strong protective immunity, live coccidiosis vaccines require the host to replicate the parasite, which has a fundamental disadvantage because it carries a high risk of causing subclinical coccidiosis and increasing the incidence of enteric bacterial infections if managed incorrectly [3,25].

The intestinal microbiome is considered the most significant microbiome in maintaining animal health. The normal gut microbiota imparts specific functions for intestinal homeostasis and host health by assisting intestinal absorption and fermentation, improving energy utilization and productivity, and resisting a variety of pathogenic microorganisms [26,27]. Several factors play a role in shaping the normal gut microbiota. Factors such as feed ingredients, supplementary feed, the farm environment, age, and genetics affect the intestinal microbial community composition [28,29,30]. In chickens, *Eimeria* infection can induce differences in the microbiota distribution and abundance compared with that of control animals, causing a gut microbial imbalance, which can affect gut microbial fermentation, host metabolism, and the PI [31,32,33].

In response to these concerns about live vaccines, this study aimed to evaluate the effects of coccidiosis vaccines in large-scale field trials at broiler farms. Approximately 900,000 chicks were vaccinated with a commercially available live coccidiosis vaccine on three open-rearing farms, where chickens are raised in the same space until they are shipped, regardless of their growth. The effectiveness of the coccidiosis vaccine was monitored using the intestinal lesion score, fecal oocyst shedding, and the PI. Concerns exist that live coccidiosis vaccines have a high potential to cause subclinical coccidiosis, an intestinal disease that can destroy the integrity of the intestinal microflora and promote the establishment and growth of potentially pathogenic bacteria. Thus, we also attempted to compare the fecal microbiota composition across farms using metagenomics.

## 2. Methods

### 2.1. Ethics Statement

All animal procedures were approved by the Institutional Animal Care and Use Committee (GNU-201116-C0084) and were conducted in accordance with the Gyeongsang National University Guidelines for the Care and Use of Experimental Animals.

### 2.2. Field Trial Design and Sample Collection

The field trial farms included selected farms with an outbreak history of coccidiosis, and several *Eimeria* species were confirmed in fecal samples using the polymerase chain reaction (PCR) and morphological methods [8]. The selected farms had an open-farm system (or full house brooding method) in which chickens were raised in the same space until they were shipped to the slaughterhouse, regardless of their growth. The three farms, with facilities of different sizes—50,000, 100,000, and 200,000 units—were located in the same area and managed by the same company (Table 1). The commercial Fortegra coccidiosis vaccine was provided free of charge by MSD Animal Health Korea Ltd. Coccidiosis vaccine was sprayed coarsely on 1-day-old chicks at the hatchery using a suitable device according to the manufacturer’s instructions. Vaccinated chicks were transported to the trial farms in approximately two hours by a special vehicle. The Fortegra vaccine contained four species of *Eimeria,* including *E. acervulina, E. maxima, E. tenella,* and *E. mivati*. Intestinal lesions (*n* = 10) and fecal oocyst shedding (*n* = 9–10) were monitored for 31 days after vaccination, including days 7, 10, 13, 16, 19, 22, 25, 28, and 31 post vaccination (Figure 1). The trial was repeated 3 times on each farm, with a total of approximately 900,000 chickens (mainly Indian River) ultimately vaccinated.

### 2.3. Evaluation of Fecal Oocyst Shedding, Lesion Scores, and the PI

Each stool sample contains mixed fresh feces obtained from multiple birds. Collected samples were sent to the laboratory in sealable plastic bags or screw cap containers. A stool sample of 1 g was mixed with 9 mL of saturated NaCl solution and thoroughly homogenized using a vortex mixer. By using a McMaster counting chamber, the number of oocysts in fecal samples was expressed as the average of two counts per sample. The number of oocysts was expressed per 1 g of feces. Based on a scoring technique according to the previously described method of Johnson and Reid [34], intestinal lesion scores at each time point were assessed. Each chicken was given an intestinal lesion score on a scale from 0 to 4, scored blindly by 2 or 3 individuals. The PI was calculated as follows: [liveability (%) × average gain (kg)/broiler age (day) × feed conversion ratio] × 100.

### 2.4. DNA Preparation and Sequencing

The collected samples, consisting of fresh feces pools collected from multiple chickens, were stored at −80 °C until microbial DNA extraction and sequencing. Total metagenomic DNA from each stool sample was obtained using a stool DNA extraction kit (MO BIO, Carlsbad, CA, USA) according to the manufacturer’s protocol. The extracted DNA concentration and amount were calculated with a QuantiFluor dsDNA system (Promega, Madison, WI, USA) and Victor Nivo Multimode Microplate Reader (PerkinElmer, Waltham, MA, USA). Based on 16S Metagenomic Sequencing Library Preparation (Illumina, San Diego, CA, USA), the hypervariable V3–V4 area was amplified from the bacterial 16S ribosomal DNA (rDNA) gene sequence using the polymerase chain reaction (PCR) with Herculase II Fusion DNA polymerase and a Nextera XT DNA index V2 kit with an Illumina adapter containing Bakt 341F/805R primers. PCR was operated under the following conditions: 3 min at 95 °C for heat denaturation, 25 cycles of 30 s at 95 °C, 30 s at 55 °C, and 30 s at 72 °C. The final 5 min extension step at 72 °C followed the completion of the last PCR cycle. Sequencing of the prepared amplicons was performed using the Illumina MiSeq platform and a 301 bp paired-end format (Macrogen, Seoul, Republic of Korea).

### 2.5. Bioinformatics and Sequencing Data Processing

The Divisive Amplicon Denoising Algorithm (DADA2) was used to process the 16S amplicon sequences. Cutadapt (v3.2) was performed to remove useless sequences including primers, adaptors and poly-A-tails from high-throughput sequencing reads. Read error correction, merging, chimera removal, and denoising processes were performed using DADA2 (v1.18.0) to sequentially obtain amplicon sequence variant (ASV) sequences. Each ASV was aligned to the organism with the highest similarity in the NCBI 16S microbial database using BLAST plus (v2.9.0). The α- and β-diversities and taxonomic assignments were performed using amplicon sequence variant (ASV) sequences. Alpha diversity is a measure of species richness and/or evenness within each sample and was evaluated with five metrics. Observed species (ASVs), Chao1 index, whole-tree *phylogenetic diversity* (PD), and Simpson and Shannon values were calculated using QIIME software (v1.9.0). To measure the distances between sample bacterial compositions, β-diversity was estimated based on the Bray–Curtis distance, and distances between groups and clustering of samples with similar microbiota were visualized using principal coordinate analysis (PCoA). To test the statistical significance of groups based on UniFac distance, nonparametric permutational multivariate analysis of variance (PERMANOVA) was assessed with R software (v3.6.2). Venn diagram analysis was performed across groups at all genus levels and visualized using Python (v3.6.2). Cladograms were constructed and linear discriminant analysis (LDA) coupled with the effect size (LEfSe) algorithm (http://huttenhower.sph.harvard.edu/, accessed on 26 April 2024) was used to identify significant among-farm microbial differences.

### 2.6. Prediction and Comparison of Functional Capacities of Fecal Microbiomes across Farms

To predict the functional abundance of bacterial community present in the three different farm feces, the Phylogenetic Investigation of Communities by Reconstruction of Unobserved States (PICRUST2; https://github.com/picrust/picrust2, accessed on 26 April 2024; v2.5.2) program [35] was used after LEfSe. Functional genes associated with bacterial invasion of epithelial cells were categorized into Kyoto Encyclopedia of Genes and Genomes (KEGG) pathways. By summing the abundances of functional genes that were annotated to the functional subsystem, the relative abundance of each subclass term of the KEGG pathways was calculated.

### 2.7. Statistical Analysis

The statistical software packages R (v3.6.2) and Instat (GraphPad, Boston, MA, USA) were used to data analysis. Data analyses of fecal oocyst production and production index (PI) were performed using one-way ANOVA followed by the Tukey–Kramer multiple comparison test. A nonparametric statistical test using the Kruskal–Wallis test was used to analyze the intestinal lesion scores and microbial abundance, and means were compared using Dunn’s multiple comparison test. Data were expressed as mean ± standard deviation values. Differences were considered statistically significant at *p* < 0.05.

## 3. Results

### 3.1. Monitoring of the Lesion Score after Vaccination

Because the commercial vaccine used in this study contained four *Eimeria* species*, E. acervulina, E. maxima, E. tenella*, and *E. mivati*, lesion scores were confirmed in three different areas of the gastrointestinal tract of chickens. The average lesion scores of the three trials (*n* = 240 to 270 animals/farm) were 0.36 to 0.64 in the duodenal region, 0.30 to 0.39 in the jejuno-ileal region, and 0.18 to 0.39 in the ceca (Table 2). For the duodenum, the JH farm (0.64 ± 0.057) showed a significantly higher lesion score (*p* < 0.0001) than both the J (0.38 ± 0.047) and S (0.36 ± 0.039) farms. Although not statistically significant, the S farm (0.39 ± 0.056) had a higher lesion score (*p* > 0.065) than the J (0.19 ± 0.035) and JH (0.18 ± 0.021) farms for the cecum. However, no difference was observed among farms for the jejunum and ileum (Table 2).

As shown in Figure 2 and Appendix A, lesion scores were analyzed in detail according to each trial, intestinal region, and sample collection day. A total of 7 out of approximately 81 time points had a mean duodenal lesion score exceeding 1. The J farm score was 1.33 ± 0.42 on day 28 in the first trial (Figure 2A) and 1.30 ± 0.6 on day 25 in the third trial (Figure 2C). The S farm score was 1.4 ± 0.34 on day 19 in the third trial (Figure 2F). The JH farm score was 1.28 ± 0.45 on day 19, 1.3 ± 0.24 on day 22, and 1.5 ± 0.5 on day 25 in the second trial (Figure 2H) and 1.15 ± 0.33 on day 10 in the third trial (Figure 2I). In the case of the jejuno-ileal region, only 1 out of approximately 81 time points exhibited a lesion score of more than 1, which was on day 25 (1.1 ± 0.35) in the third trial (J Farm, Figure 2C). For the cecum, only 4 out of approximately 81 time points exhibited a mean lesion score exceeding 1. The J farm had a score of 1.1 ± 0.43 on day 25 in the third trial (Figure 2C). The S farm had a score of 1 ± 0.49 on day 28 in the second trial (Figure 2E), 1.3 ± 0.5 on day 13, and 1.4 ± 0.52 on day 16 in the third trial (Figure 2F). For the JH farm, cecal lesion scores averaged less than 1 at all tested time points.

### 3.2. Analysis of Fecal Oocyst Shedding after Vaccination

The numbers of fecal oocysts per gram were calculated using a pool of fresh feces from several chickens. The results obtained from three trials (*n* = 240 to 270 samples/farm) showed significant differences in the number of oocysts among farms, with an average of 100,100 ± 202,559 oocysts for the J farm, 61,630 ± 142,347 for the S farm, and 19,766 ± 50,669 for the JH farm (Table 3). Fecal oocysts were mainly detected from day 13 after vaccination, showing large variations depending on the farm and the building within the farm. Additionally, each farm exhibited specific oocyst production patterns (Figure 3).

### 3.3. PI Increases after Vaccination

The PI was calculated using the feed conversion rate, weight gain, age, and liveability. Compared with the average PI of each farm’s previous 9 to 10 trials, the J farm’s PI increased by 8.22 (*p* < 0.604, 342.33 ± 13.28 vs. 334.11 ± 24.92), the S farm’s PI increased by 23.55 (*p* < 0.355, 327.33 ± 22.59 vs. 303.78 ± 39.20), and the JH farm’s PI increased by 2.47 (*p* < 0.890, 318.67 ± 18.01 vs. 316.20 ± 37.10), although there was no difference in the statistical analysis (Table 4). The average PI of the integration over the 33 months from January 2021 to September 2023 was 318.95 ± 14.48 (Table 4).

### 3.4. Analysis of Fecal Microbiota Composition across Farms

#### 3.4.1. General Information and Sample Sequencing

Because *Eimeria* infection resulted in changes in the microbial abundance and community of the gut or feces, the microbiota in fecal samples collected from three vaccinated broiler farms were analyzed using 16S rRNA microbial profiling. The detailed sequencing data for each sample are presented in Appendix A. The average number of original total reads was 127,225 (range: 102,062 to 173,334), and that of the highly qualified reads used for the final analysis was 51,098 (range: 28,194 to 64,223). A total of 473,412 ASVs were obtained from 15 samples; the average number of remaining ASVs was 31,561 (range: 18,143 to 64,233) (Appendix A). The rarefaction curve values for the samples with ASVs and the Shannon, Gini–Simpson, and whole-tree PD analyses indicated sufficient data sampling and adequate sequencing depth, and the 16S rRNA gene-based sequence database covered most microbial communities (Appendix A).

#### 3.4.2. Diversity Analyses and Gut Microbial Community

The α-diversity represents species richness, evenness, and diversity. Microbial diversity between the J and JH farms showed significant variation based on the whole-tree PD (*p* < 0.05) and Simpson (*p* < 0.05) index results (Figure 4C,D). However, the ASVs and the Chao1 and Shannon index results indicated no significant between-group differences (Figure 4A,B,E). Principal coordinate analysis with the taxonomy abundance of species was performed using the Bray–Curtis distance statistic. Principal component 1 and principal component 2 explained 34.53% and 20.69%, respectively, of the variance of variables, and the cumulative contribution rate was 55.22%. The resulting plot revealed that β-diversity significantly differed among the three farms (PERMANOVA, pseudo-F = 4.606, *p* = 0.001) (Figure 4F). Additionally, there were significant differences in β-diversity between the J and JH farms (pseudo-F = 4.863, *p* = 0.009), S and JH farms (pseudo-F = 4.621, *p* = 0.007), and J and S farms (pseudo-F = 4.158, *p* = 0.011) (Appendix A).

#### 3.4.3. Analysis of Microbial Abundance and Composition

To characterize the microbial taxonomic abundance ratios in broiler farms, we analyzed the fecal microbiota at the phylum and genus levels based on the Kruskal–Wallis H test result (*p* < 0.05) (Table 5). At the phylum level, Bacillota, Actinomycetota, Pseudomonadota, and Bacteroidota were most dominant. The average abundances of Bacillota, Actinomycetota, Pseudomonadota, and Bacteroidota were 79.4% (range: 62.6% to 97.0%), 16.4% (range: 0.84% to 36.2%), 2.4% (range: 0.2% to 9.2%), and 1.6% (range: 0% to 12.7%), respectively. Significant differences were found for Bacillota and Actinomycetota, especially between the J and JH farms. At the genus level, *Lactobacillus*, *Ligilactobacillus*, *Limosilactobacillus*, *Corynebacterium*, and *Brevibacterium* were most abundant. The average abundances of *Lactobacillus*, *Ligilactobacillus*, *Limosilactobacillus*, *Corynebacterium*, and *Brevibacterium* were 25.1% (range: 8.5% to 49.4%), 22.1% (range: 3.6% to 58.4%), 10.9% (range: 1.7% to 43.8%), 9.2% (range: 0% to 21.7%), and 4.9% (range: 0.2% to 15.3%), respectively. Of the top 20 genera analyzed, 10 (*Corynebacterium*, *Brevibacterium*, *Atopostipes*, *Romboutsia*, *Brachybacterium*, *Clostridium*, *Tidjanibacter*, *Mammaliicoccus*, *Jeotgalicoccus*, and *Ruminiclostridium*) were significant in terms of their relative abundance. Eight of these genera (all except *Atopostipes* and *Romboutsia*) were significantly different between the J and JH farms (Table 5).

To compare the microbial composition similarity among the three different farms, Venn diagram analyses at the genus level were performed (Figure 5A, Appendix A). All three farms shared 61 of the 244 genera. Overall, JH, J, and S farms had 116, 2, and 21 unique genera, respectively (Appendix A). Eighteen (all except *Ruminiclostridium* and *Lysobacter*) of the 20 most abundant genera were shared by all three farms. The genus *Lysobacter* was unique to the JH farm. Farms J and S had no unique genera among the 20 most abundant genera (Appendix A). To identify significant differential microbial taxa using abundance values across different broiler farms (Figure 5B,C), LEfSe was performed to produce a cladogram and a histogram of LDA scores. An LDA effect size of >4 and a significance value of <0.05 were used as thresholds for LEfSe. Taxa with significant differential abundances were detected using nonparametric factorial Kruskal–Wallis tests. In total, 31 differential microbial enrichments were found across all three farms: 19 from the J farm, 4 from the S farm, and 8 from the JH farm. The major species populations were *Corynebacterium casei* and *Brevibacterium avium* on the J farm, *Lactobacillus crispatus* and *Tidjanibacter massiliensis* on the JH farm, and *Streptococcus alactolyticus* and *Romboutsia timonensis* on the S farm. These different abundances occurred mainly in the phyla Bacillota and Actinomycetota.

To more accurately reflect the physiological consequences of the microbiome profile in feces, functional abundance was inferred using LEfSe and PICRUSt2. In total, 17 distinct features were found with a cut-off log-LDA score of 3 (Figure 5D). Most of the aspects over-represented at the J farm were related to cell structure biosynthesis, amino acid biosynthesis, the generation of precursor metabolites and energy, and vitamin synthesis. Four and two MetaCyc functional pathways involved in fatty acid and lipid biosynthesis, amino acid biosynthesis, and carbohydrate degradation were upregulated at the JH farm and S farm, respectively (Appendix A). These different functional pathways may have the potential to cause a difference in the PI between the J and JH farms, although no difference was found (*p* < 0.14) (Table 4). One concern regarding *Eimeria* infection is related to pathogenic bacteria present in feces obtained from vaccinated chickens. However, the KEGG pathway analysis predicted that bacterial invasion of the epithelial cell pathway was not significant across farms (Figure 5E).

## 4. Discussion

Avian coccidiosis in poultry farms causes serious economic losses in the poultry industry due to reduced feed efficiency, weight loss, and increased mortality [3,9]. It also alters the microbial balance, which can affect the composition of microbial communities in the gut and litter [10,36,37]. Thus, commercial poultry feces are an important issue affecting performance and gut health, especially when recycling bedding. Live coccidia spray vaccines, which are currently mainly used instead of anticoccidial drugs to prevent damage caused by coccidia outbreaks, require the host to replicate the parasite to induce strong protective immunity. This process has the fundamental disadvantage of having a high risk of causing asymptomatic coccidiosis and, if managed incorrectly, increasing the incidence of intestinal bacterial infections [3,18,25]. Therefore, the impact of the live coccidiosis vaccine was monitored through a large-scale field trial in broiler farms using three parameters (body weight gain, intestinal lesion score, and fecal oocyst shedding) as indicators to measure the degree of coccidial infection, and the fecal microbial composition was compared across trial farms.

The most important parameter in broiler chickens infected with *Eimeria* spp. is the economic loss due to weight loss. There are many previous reports of a negative relationship between *Eimeria* infection and body weight gain [9,20,36,38,39,40,41,42,43]. Chickens infected with 10,000 and 100,000 *E*. *acervulina* oocysts weighed approximately 11% and 19.8% less than uninfected control chickens, respectively [20,38]. Broilers with 10,000 *E. maxima* oocysts showed a significant reduction in body weight gain compared with uninfected control chickens [9,39,40]. Additionally, broilers infected with 50,000 *E. maxima* oocysts weighed approximately 38% less than uninfected control chickens [41]. In broiler chickens infected with 140,000 *E. maxima* oocysts, body weight was reduced by approximately 11.6 to 22.8% depending on the isolate [42]. Chickens infected with 20,000 *E. tenella* oocysts weighed 17.6% less than uninfected chickens [36]. However, no significant difference in body weight gain was observed in broilers infected with *E. tenella* at four doses: 6250, 15,500, 25,000, and 50,000 oocysts [43]. Collectively, these results indicate that *Eimeria*-infected chickens lost more than 10% of their body weight under experimental conditions, although the rate of weight loss varied depending on age, dose, sex, *Eimeria* strain, and host strain [9]. In the present nine trials across three broiler farms, the PI after *Eimeria* vaccination increased by 2.45 to 23.55% compared with the PIs of the previous 9 to 10 trials for each farm. These results demonstrate that the application of a commercial coccidiosis vaccine has a positive impact on economic value and can reduce public concerns by reducing the use of anti-coccidiosis drugs to prevent the development of coccidiosis.

The fecal oocyst number and intestinal lesion score are also used as anticoccidial indices in many previous experiments. Although oocyst production is more variable and less correlated than lesion scoring and weight gain, chickens infected with more oocysts generally have higher lesion scores or morbidity [9,44,45], indicating that the number of oocysts in feces or litter is an important factor in subsequent infection. In this study, the average number of fecal oocysts ranged from 19,766 to 100,100, with high variability depending on the sampling time, farm, and even the building on the farm. The period when the most oocysts were released was around the second half of the third to fourth week after vaccination, and then the fecal oocyst shedding rate suddenly decreased, demonstrating that immunity due to the vaccine was complete. The vaccine trials were performed in open chicken coops. This system does not gradually expand the space according to the growth of the chickens. Therefore, the opportunity for vaccinated chickens to ingest fecal oocysts produced in the first and second replications will be reduced in an open chicken coop compared with that in a system that expands the breeding space according to the growth of broilers. This open system may slightly delay the development of immunity to the vaccine, although additional clarification and investigations are needed.

Although the number of fecal oocysts per gram varied across the chicken farms, Lee et al. reported that the mean oocyst number per gram of positive samples collected from field farms (*n* = 280) was 20,810 [7]. The results of an epidemiological investigation of 278 broiler farms in Korea showed that more than 10,000 oocysts per gram were detected in 36.6% of fecal samples [8]. In the current study, the average number of oocysts in fecal samples was 19,766 for the JH farm, 61,630 for the S farm, and 100,100 for the J farm. Compared with the results of our previous investigations [7,8], it can be inferred that greater numbers of oocysts were detected at the S and J farms. This difference in the number of oocysts in fecal samples may be due to the lack of anticoccidial agent use during vaccination, which allows the oocysts to replicate easily in the host’s intestinal epithelial cells. In addition, the vaccine used in this study included four *Eimeria* species: *E. acervulina*, *E. maxima*, *E. tenella*, and *E. mivati.* Of these, *E. acervulina* and *E. mivati* generally produce more oocysts, which is considered another reason for the increased number of oocysts. In this study, the mean lesion scores of the farms ranged from 0.36 to 0.64 in the duodenal section, from 0.30 to 0.39 in the jejuno-ileal section, and from 0.18 to 0.39 in the cecal section. Given that the average lesion score was 0.25 in control chickens without *Eimeria* infection [41], it is believed that the vaccine had no or little effect on the lesion scores.

*Eimeria* infection, especially *E. tenella*, can cause an imbalance in the gut microbiota composition [46,47], thus altering the microbial abundance and community in feces or manure. Although vaccination in this trial had little or no effect on lesion scores, there is a potential for microbial alteration because the vaccine contained *E. tenella*. Therefore, a comparative analysis of the fecal microbiota obtained from vaccinated farms was performed using 16S rRNA microbial profiling. In the current study, Bacillota (syn, Firmicutes) and Actinomycetota (syn, Actinobacteria) were the most dominant phyla. The abundances of Bacillota and Actinomycetota were 68.3% and 30% for the J farm, 80.3% and 15.8% for the S farm, and 89.6% and 3.5% for the JH farm. Considering that the taxonomy abundance ratios of phyla differed and β-diversity was significantly different across all vaccinated farms, microbial abundance ratios were expected to vary significantly across farm feces. Similarly, the predominant phyla are highly diverse among previous studies. Several scientists analyzed and reported on the composition of the microbiome of chicken feces [26,48,49]. Feces from two broiler lines derived from Arbor Acres, namely fatty (FL) and lean (LL) chickens, were collected between 37 and 40 weeks of age [26]. The most dominant phyla were Firmicutes and Bacteroidetes (syn, Bacteroidota)*,* which accounted for 53.44% and 41.09% of the phyla in LL and 71.36% and 23.40% of the phyla in FL, respectively. The proportion of Bacteroidetes was significantly different between the LL and FL samples (LL > FL, 41.09% vs. 23.40%, *p* = 0.034) [26]. Firmicutes (55.9%), Bacteroidetes (35.5%), Proteobacteria (syn, Pseudomonadota) (1.7%), and Actinobacteria (0.1%) were the dominant taxonomic phyla in the fecal bacterial community of healthy 18-week-old white leghorn chickens [48]. Proteobacteria (81%), Firmicutes (11%), Bacteroidetes (2.7%), and Actinobacteria (1.3%) were the predominant taxonomic phyla in the fecal microbiota of broiler chickens [49]. Together, these results suggest the possibility that between-farm or between-line differences in fecal microbial taxonomic abundance ratios may be more important than changes in the fecal microbial composition due to *Eimeria* vaccination.

To reduce agricultural waste and increase farm profits, broiler farms recycle the used manure. Zheng et al. [29] reported that the composition of microorganisms present in the manure is important for the composition of the gut microbiota during the initial colonization period and may influence the health and growth of chickens [29]. Opportunistic pathogenic bacteria in the gut microbiota, such as members of the genera *Clostridium*, *Lysinibacillus*, and *Escherichia,* were clearly increased both in *E. tenella*-infected Arbor Acres broilers and specific pathogen-free white leghorn chickens [50]. One concern with *Eimeria* infection is related to pathogenic bacteria present in feces obtained from vaccinated chickens. In this study, KEGG pathway analysis predicted that bacterial invasion of the epithelial cell pathway was not significant across farms. Although the microbial community composition of litter samples is significantly different from that of cloacal samples [51,52], this result suggests that *Eimeria* vaccination has little or no effect on the abundance rate of pathogenic bacteria. Additional clarification and investigations are needed to verify this finding.

## 5. Conclusions

Large-scale field trials showed that a live coccidiosis vaccine plays a positive role in increasing the PI, reducing the use of anticoccidials, and saving farmers’ labor with little or no alteration of the fecal microbiota composition, resulting in increased economic value. Although there are limitations because the data were obtained by directly comparing the results of broiler farms with different environments, including different bedding conditions, it is believed that the use of coccidiosis vaccines as an alternative to the use of prophylactic anticoccidials can improve poultry health and management in the poultry industry. Collectively, these findings could not only serve as a roadmap for understanding coccidia vaccination under field conditions and developing immunoprotective approaches against this disease, but can also provide an opportunity to understand avian protozoan diseases.

## Figures and Tables

**Figure 1 vaccines-12-00800-f001:**
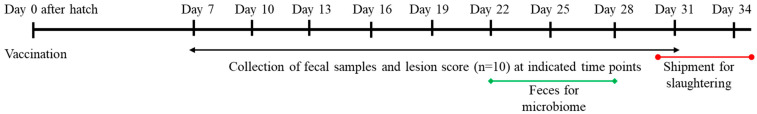
Schematic outline of the trial design. One-day-old chicks were inoculated with a commercially available coccidiosis vaccine by coarse spray using an automatic device at a hatchery facility and delivered to trial farms using a specialized vehicle. The chicks were raised in an open breeding system from day 1 until slaughter. Samples for fecal oocyst counts (*n* = 9 to 10) and lesion scores (*n* = 10) were obtained on days 7, 10, 13, 16, 19, 22, 25, 28, and 31 post vaccination.

**Figure 2 vaccines-12-00800-f002:**
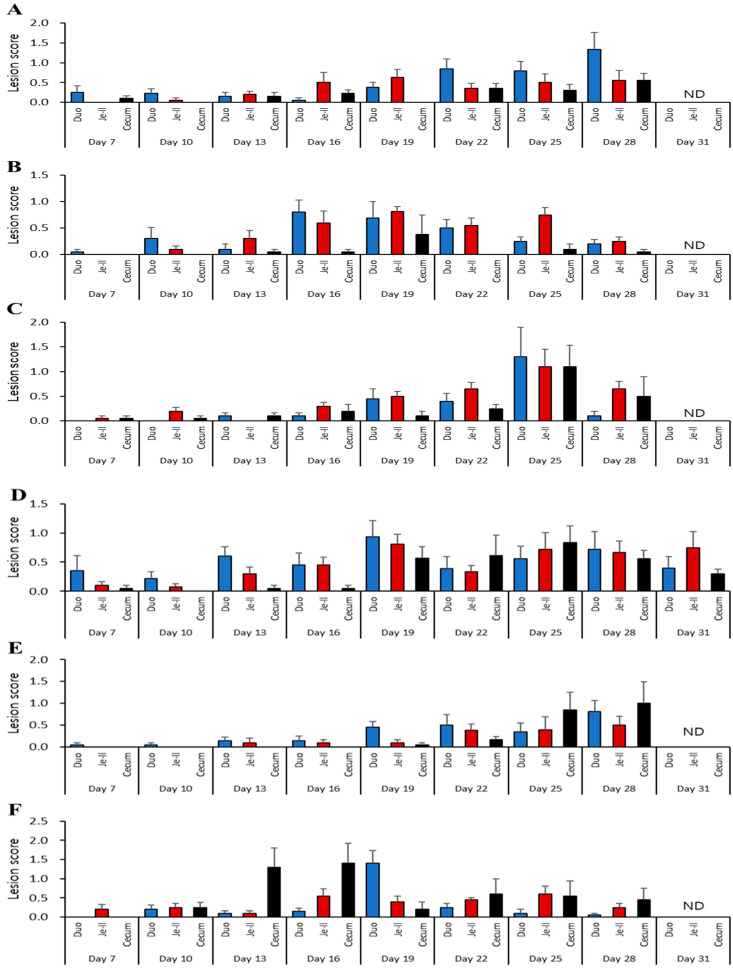
Monitoring of intestinal lesion scores after coccidiosis vaccination in 1-day-old chicks. Lesions were scored on a scale of 0 to 4, with 0 representing no gross lesions and 4 representing extensive and severe hemorrhage or lesions (depending on *Eimeria* species). Data from the J farm (**A**–**C**), S farm (**D**–**F**), and JH farm (**G**–**I**) are shown. Results from the first trial (**A**,**D**,**G**), second trial (**B**,**E**,**H**), and third trial (**C**,**F**,**I**) are also shown. Bars (*n* = 10) represent the mean ± standard error. Duo, duodenum; Je-Il, jejunum to ileum; J, Junho farm; S, Saemgol farm; JH, Jeonghwa farm; ND, not determined.

**Figure 3 vaccines-12-00800-f003:**
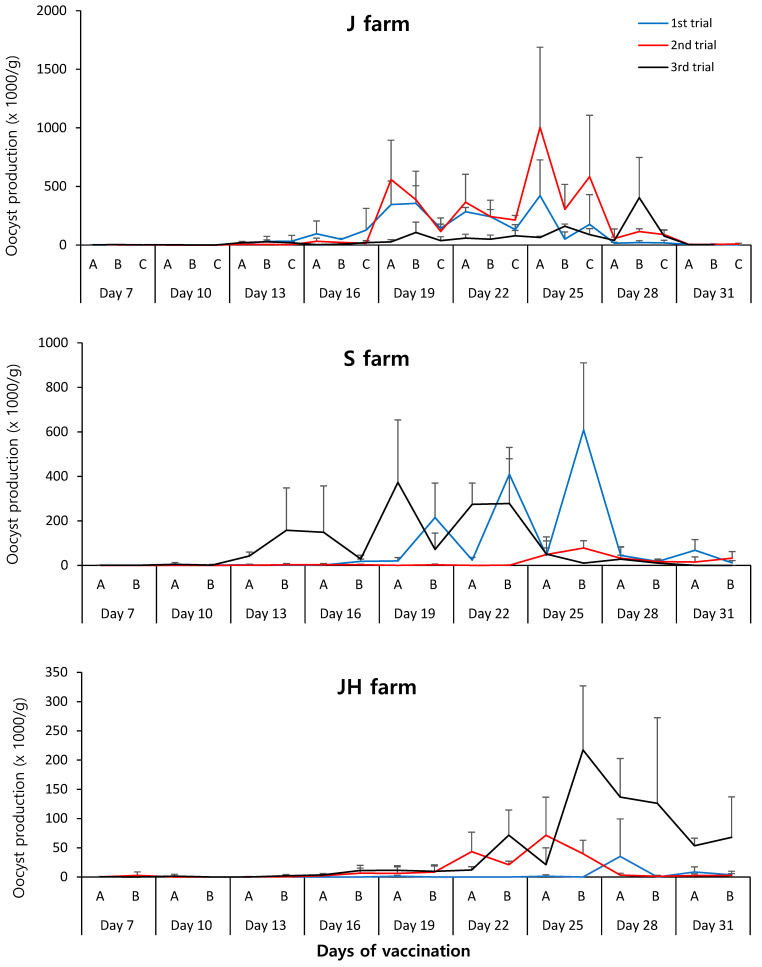
Monitoring of fecal oocyst shedding after coccidiosis vaccination in 1-day-old chicks. Each fecal sample consisted of a pool of fresh manure collected from several chickens. Oocyst numbers per gram were calculated using a McMaster counting chamber. Data represent the mean ± standard deviation. A, B, and C in each figure represent buildings on the farms. J, Junho farm; S, Saemgol farm; JH, Jeonghwa farm.

**Figure 4 vaccines-12-00800-f004:**
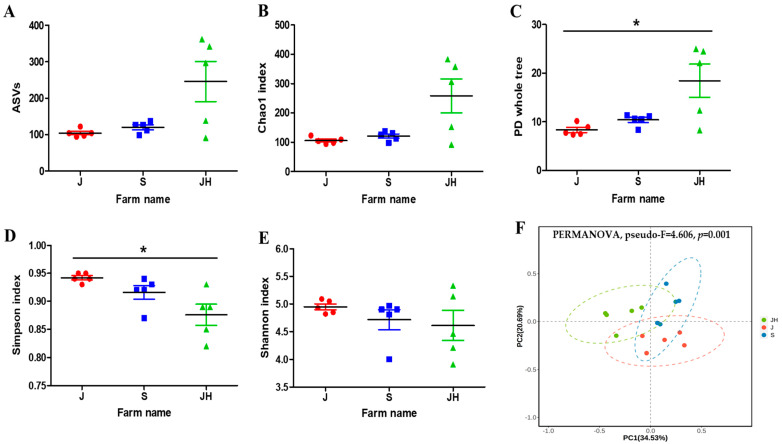
Diversity analysis of fecal samples collected from broiler farms. (**A**) ASVs. (**B**) Chao1 index. (**C**) Whole-tree phylogenetic diversity. (**D**) Gini–Simpson index. (**E**) Shannon index. The Kruskal–Wallis test was employed to assess significant differences between groups. * *p <* 0.05 indicates a statistically significant difference. (**F**) Beta diversity was visualized by MDS based on the Bray–Curtis dissimilarity. The *p*-value was derived from a PERMANOVA test with 999 permutations. ASVs, amplicon sequence variants. J, Junho farm; S, Saemgol farm; JH, Jeonghwa farm.

**Figure 5 vaccines-12-00800-f005:**
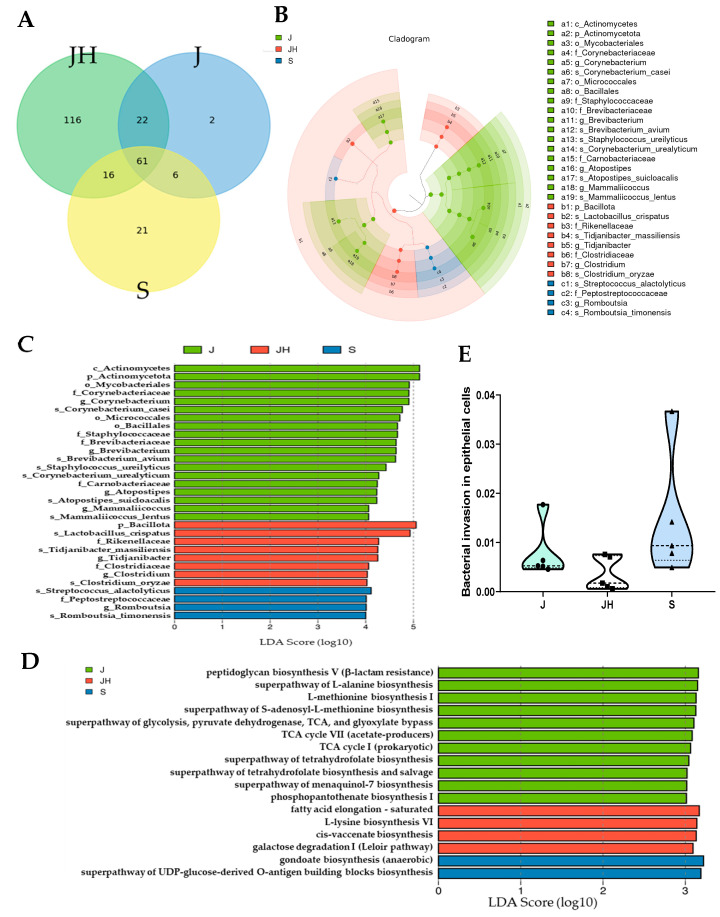
Differences in abundances, predicted pathways, and invasion abundance of bacterial communities in vaccinated farm feces. (**A**) Venn diagram showing the ASV numbers of unique and shared genera. (**B**,**C**) Cladogram (**B**) and LDA (**C**) of LEfSe. Histograms of the LDA scores were computed for differentially abundant bacterial taxa across farms. A significance value of less than 0.05 and an LDA effect size of greater than 4 were used as thresholds for the LEfSe. (**D**) Predicted MetaCyc pathway abundance. A significance value of less than 0.05 and an LDA effect size of greater than 3 were used as thresholds for the LEfSe. (**E**) The abundance of bacterial invasion of the epithelial cell pathway with KEGG pathway level 3 analysis. LEfSe, linear discriminant analysis coupled with the effect size; LDA, linear discriminant analysis; ASV, amplicon sequence variant; KEGG, *Kyoto Encyclopedia of Genes and Genomes*; J, Junho farm; S, Saemgol farm; JH, Jeonghwa farm.

**Table 1 vaccines-12-00800-t001:** Information on broiler farms included in field trials.

Name of Farm	Trial Period	Capacity (Birds)	Farming Style	Bedding Status	No. of Buildings	Location
J	May 2023–October 2023	50,000	Open breeding system *	Recycled	3	Jinangun, Jeollabuk-do
S	May 2023–August 2023	100,000		New	2	
JH	September 2022–May 2023	200,000		Recycled	8	

* An open breeding system refers to a farm where chickens are raised in the same space until they are shipped, regardless of their growth. J, Junho farm; S, Saemgol farm; JH, Jeonghwa farm.

**Table 2 vaccines-12-00800-t002:** Lesion scores after coccidiosis vaccination in 1-day-old chicks.

Location		J	S	JH	*p*-Value
Lesion score	Duodenum	0.38 ± 0.047 ^a^	0.36 ± 0.039 ^a^	0.64 ± 0.057 ^b^	0.0001 *
	Jejunum-ileum	0.39 ± 0.035	0.34 ± 0.033	0.30 ± 0.029	0.1811
	Cecum	0.19 ± 0.035	0.39 ± 0.056	0.18 ± 0.021	0.0655

The Kruskal–Wallis H test was used to assess significant differences (* *p* < 0.05). Data (*n* = 240 to 270) represent the mean ± standard error of the three trials. Different letters within the same row indicate statistically significant differences (* *p* < 0.05). J, Junho farm; S, Saemgol farm; JH, Jeonghwa farm.

**Table 3 vaccines-12-00800-t003:** Fecal oocyst shedding after coccidiosis vaccination in 1-day-old chicks.

Name of Farm	J	S	JH	*p*-Value
No. of oocysts	100,100 ± 202,559 ^a^	61,630 ± 142,347 ^b^	19,766 ± 50,669 ^c^	0.0001 *

Data were analyzed using one-way ANOVA followed by the Tukey–Kramer multiple comparison test. Data (*n* = 240 to 270) represent the mean ± standard deviation of the three trials. Different letters within the same row indicate statistically significant differences (* *p* < 0.05). Fecal oocyst counts refer to the number per gram. J, Junho farm; S, Saemgol farm; JH, Jeonghwa farm.

**Table 4 vaccines-12-00800-t004:** Production index after coccidiosis vaccination.

Name of Farm	Production Index		*p*-Value
	Before Vaccine (*n* = 9–10)	After Vaccine (*n* = 3)	
Integration	318.95 ±14.48 *	ND	
J farm	334.11 ± 24.92	342.33 ± 13.28	0.604
S farm	303.78 ± 39.20	327.33 ± 22.59	0.355
JH farm	316.20 ± 37.10	318.67 ± 18.01	0.890

* Production index for 33 months from January 2021 to September 2023. The production indices of farms J, S, and JH before vaccination are the averages of 9, 9, and 10 trials, respectively. Data represent the mean ± standard deviation. Statistical analysis was performed using Student’s *t*-test. J, Junho farm; S, Saemgol farm; JH, Jeonghwa farm; ND, not determined.

**Table 5 vaccines-12-00800-t005:** Taxonomy abundance ratios of phyla and the top 20 genera.

Taxonomy		J Farm (%)	S Farm (%)	JH Farm (%)	*p*-Value
Phylum	Bacillota	68.3 ± 7.9 ^a^	80.3 ± 4.7 ^a,b^	89.6 ± 7.6 ^b^	0.0068 *
	Actinomycetota	30.0 ± 7.5 ^a^	15.8 ± 6.6 ^a,b^	3.5 ± 3.4 ^b^	0.0044 *
	Pseudomonadota	1.6 ± 1.1	3.5 ± 3.3	2.2 ± 2.8	0.3097
	Bacteroidota	0.1 ± 0.1	0.3 ± 0.3	4.4 ± 5.2	0.1079
Genus	*Lactobacillus*	21.1 ± 10.6	18.2 ± 6.0	36.0 ± 11.0	0.0554
	*Ligilactobacillus*	12.9 ± 5.2	36.3 ± 16.2	17.1 ± 15.5	0.0626
	*Limosilactobacillus*	11.1 ± 5.5	6.7 ± 3.5	14.9 ± 16.4	0.4163
	*Corynebacterium*	16.3 ± 5.4 ^a^	11.1 ± 5.2 ^a,b^	0.3 ± 0.4 ^b^	0.0006 *
	*Brevibacterium*	10.2 ± 4.0 ^a^	2.8 ± 1.1 ^a,b^	1.8 ± 2.6 ^b^	0.0018 *
	*Staphylococcus*	6.0 ± 5.7	4.0 ± 2.0	1.1 ± 1.5	0.0662
	*Escherichia*	1.6 ± 1.2	3.4 ± 3.4	0.9 ± 0.9	0.1646
	*Weissella*	1.4 ± 1.7	2.5 ± 2.6	1.8 ± 1.7	0.8580
	*Streptococcus*	2.5 ± 2.6	2.8 ± 3.8	0.2 ± 0.2	0.1953
	*Enterococcus*	1.4 ± 1.0	2.4 ± 0.9	1.5 ± 2.1	0.1322
	*Atopostipes*	4.1 ± 4.8 ^a^	0.3 ± 0.2 ^b^	0.6 ± 0.8 ^a,b^	0.0239 *
	*Romboutsia*	1.5 ± 1.3 ^a,b^	2.6 ± 0.8 ^a^	0.4 ± 0.5 ^b^	0.0042 *
	*Brachybacterium*	2.5 ± 0.7 ^a^	1.0 ± 0.4 ^a,b^	0.6 ± 0.6 ^b^	0.0009 *
	*Clostridium*	0.03 ± 0.0 ^a^	0.1 ± 0.1 ^a,b^	3.6 ± 4.4 ^b^	0.0150 *
	*Tidjanibacter*	0.01 ± 0.0 ^a^	0.05 ± 0.0 ^a,b^	3.6 ± 5.0 ^b^	0.0020 *
	*Mammaliicoccus*	2.6 ± 1.6 ^a^	0.8 ± 0.5 ^a,b^	0.3 ± 0.3 ^b^	0.0014 *
	*Faecalibacterium*	0.1 ± 0.1	0.1 ± 0.1	3.1 ± 2.9	0.0745
	*Jeotgalicoccus*	1.3 ± 0.3 ^a^	0.7 ± 0.3 ^a,b^	0.1 ± 0.2 ^b^	0.0005 *
	*Ruminiclostridium*	0.0 ± 0.0 ^a^	0.003 ± 0.0 ^a,b^	1.4 ± 1.9 ^b^	0.0215 *
	*Lysobacter*	0.0 ± 0.0	0.0 ± 0.0	1.2 ± 2.8	0.3679

Distribution and relative abundance of microbial phyla and the top 20 genera assignable to ASVs in different farm samples. The Kruskal–Wallis H test was used to assess significant differences (* *p* < 0.05). Data represent the mean ± standard deviation. (*n* = 5). Different letters within the same row indicate statistically significant differences (*p* < 0.05). ASVs, amplicon sequence variants; J, Junho farm; S, Saemgol farm; JH, Jeonghwa farm.

## Data Availability

The data presented in this study are available in the NCBI Sequence Read Archive, SAMN39939184–SAMN39939193 and SAMN40596648–SAMN40596652.

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
