# Peer review of "Large-Scale Field Trials of an Eimeria Vaccine Induce Positive Effects on the Production Index of Broilers"

_vaccines, 2024, doi:10.3390/vaccines12070800_

Round 1

Reviewer 1 Report

Comments and Suggestions for Authors

The manuscript presents a comprehensive evaluation of a commercial live coccidia vaccine's effectiveness in broiler farms. The study involves approximately 900,000 chicks across three different farms, focusing on parameters such as lesion scores, fecal oocyst shedding, and production index (PI). The findings suggest that the vaccine has a positive impact on economic value and reduces the reliance on anticoccidial drugs without significantly altering the fecal microbiota composition.

The current study compares lesion scores and production performance in three poultry farms following the application of a coccidiosis vaccine. Including a small control group of unvaccinated chickens would better support the conclusions of the study.

The term "Coccidia vaccines" should be corrected to "coccidiosis vaccines."

There is a lack of clear description regarding lesion scores and oocyst shedding after vaccination in Table 2. In other words, the basis or theoretical foundation for the author's choice to conduct a combined analysis of data from all time points after immunization is not well explained. Alternatively, the conclusion that the author intends to clarify has not been clearly described.

Figure 2 is not easy for the reader to understand.

Comments on the Quality of English Language

Minor editing of English language required

Author Response

We sincerely appreciate the editor and reviewers for their valuable time, and insightful recommendations for enhancing our manuscript. The authors have diligently considered the comments, and below are our point-for-point responses. We trust these will align with your expectations and contribute to the further improvement of our manuscript.

Reviewer’s comments 1: 

The manuscript presents a comprehensive evaluation of a commercial live coccidia vaccine's effectiveness in broiler farms. The study involves approximately 900,000 chicks across three different farms, focusing on parameters such as lesion scores, fecal oocyst shedding, and production index (PI). The findings suggest that the vaccine has a positive impact on economic value and reduces the reliance on anticoccidial drugs without significantly altering the fecal microbiota composition.

1. The current study compares lesion scores and production performance in three poultry farms following the application of a coccidiosis vaccine. Including a small control group of unvaccinated chickens would better support the conclusions of the study.

Response: When broiler chickens are raised on farms where coccidiosis occurs without anti-coccidial agents, production performance is very poor and chickens are generally vulnerable to secondary infections. As a result, farm productivity is so low that it is difficult to realistically create a control group. In additions, this work planned to set up a small control group in the same building with vaccinations, but this was very difficult considering the farmers and the slaughter process. Most processes, including loading chickens into vehicles at night, slaughtering, and weighing, are automated, making it very difficult to isolate a small control group. Therefore, the production index, which is the most important part in broiler industry, was analyzed using the latest production index owned by the same company.

2. The term "Coccidia vaccines" should be corrected to "coccidiosis vaccines."

Response: The indicated word was corrected.

3. There is a lack of clear description regarding lesion scores and oocyst shedding after vaccination in Table 2. In other words, the basis or theoretical foundation for the author's choice to conduct a combined analysis of data from all time points after immunization is not well explained. Alternatively, the conclusion that the author intends to clarify has not been clearly described.

Response: We appreciate the reviewer's comment and have made the necessary revisions. In the revised manuscript, Table 2 separated into Table 2 for lesion scores and Table 3 for fecal oocyst shedding. Tables 2 and 3, which comprehensively analyze data at all time points after vaccination, are supported by Figures 2 and 3, which display data at each time point. The conclusion was modified and rearranged.

4. Figure 2 is not easy for the reader to understand.

Response: The supplementary table 1 (Table S1) containing related data has been added to improve understanding of Figure 2.

Reviewer 2 Report

Comments and Suggestions for Authors

The manuscript "Large-scale field trials of an Eimeria vaccine induce positive 2 effects on the production index of broilers”. The study is well-structured, and the experimental design is great. However, there are several areas where the manuscript could be improved for clarity, depth, and overall scientific rigor.

The following comments need to be reviewed and addressed by the authors:

1.     Revise the manuscript to correct minor grammatical errors.

2.     Strengthen the conclusion by more explicitly linking the results to the broader implications for poultry health and management. Suggest practical applications of the findings and potential for future research.

L36: Update reference, use information from 2023-2024

L43-45: discuss where each species has an impact in the digestive system.

L57-58: revise, discuss resistance in coccidia vaccine rather than antibiotics

Table 1: discuss how bedding status has an impact on coccidia. Recycled bedding and new: this could provide a source of variation that you could not measure. May be this is the only weak point about the whole experiment, elaborate.

Fig 1 A: remove pictures.

L147-148: revise font.

Table 2 and wherever applies, define J, S, JH.

Discussion: some information was mentioned earlier in the introduction, revise.

Author Response

We sincerely appreciate the editor and reviewers for their valuable time, and insightful recommendations for enhancing our manuscript. The authors have diligently considered the comments, and below are our point-for-point responses. We trust these will align with your expectations and contribute to the further improvement of our manuscript.

Reviewer’s comments 2: 

The manuscript "Large-scale field trials of an Eimeria vaccine induce positive effects on the production index of broilers”. The study is well-structured, and the experimental design is great. However, there are several areas where the manuscript could be improved for clarity, depth, and overall scientific rigor. The following comments need to be reviewed and addressed by the authors:

1. Revise the manuscriptto correct minor grammatical errors.

Response: Grammatical errors were corrected.

2. Strengthen the conclusionby more explicitly linking the results to the broader implications for poultry health and management. Suggest practical applications of the findings and potential for future research.

Response: The additional sentence was added with “it is believed that the use of coccidiosis vaccines as an alternative to the use of prophylactic anticoccidials can improve poultry health and management in the poultry industry” in line 468.

3. L36: Update reference, use information from 2023-2024

Response: The modified sentence was added as “The estimated global cost of coccidiosis control, based on 2016 prices, was £10.36 billion [1] and the financial cost of coccidiosis per chicken raised in 2022 was £0.30 [2]” in line 37. The related reference [2] published in 2024 was added in the section of reference.

 4. L43-45: discuss where each species has an impact in the digestive system.

Response: Typically, Eimeria spp. infections present similar clinical signs, such as diarrhea, and reduced feed conversion and weight gain. However, infected chickens showed a variety of clinical signs depending on the infectious dose of sporulated oocysts. Thus, the sentence was modified as “eliciting a variety of clinical signs in infected chickens depending on the infectious dose of sporulated oocysts” in line 46.

5. L57-58: revise, discuss resistance in coccidia vaccine rather than antibiotics

Response: The additional sentence was added as “Of them, vaccines that do not cause any resistance are receiving great attention in the chicken industry to prevent coccidiosis” in line 62.

 6. Table 1: discuss how bedding status has an impact on coccidia. Recycled bedding and new: this could provide a source of variation that you could not measure. May be this is the only weak point about the whole experiment, elaborate.

Response: We appreciate the reviewer's comments and would like to provide a revised statement. In South Korea, many farms reused bedding multiple times to save money and reduce agricultural waste. However, there is no information yet on whether recycled or new bedding affects coccidiosis. Although this experiment included two conditions of bedding, it was difficult to obtain accurate information about how bedding conditions affect the occurrence of coccidia. Thus, a modified sentence was added as “Although there are limitations because the data were obtained by directly comparing the results of broiler farms in different environments including different bedding conditions” in the section of conclusion in line 467.

7. Fig 1 A: remove pictures.

Response: Figure 1A was removed.

8. L147-148: revise font.

Response: The indicated error was corrected in line 150.

9. Table 2 and wherever applies, define J, S, JH.

Response: “J, Junho farm; S, Saemgol farm; JH, Jeonghwa farm” was added in the footnotes of Tables and Figures.

10. Discussion: some information was mentioned earlier in the introduction, revise.

Response: The revised sentence was added with “Avian coccidiosis in poultry farms causes serious economic losses in the poultry industry due to reduced feed efficiency, weight loss, and increased mortality” in line 355.

Reviewer 3 Report

Comments and Suggestions for Authors

While the article is interesting, there are errors in the results analysis section that need to be resolved. In addition, some specific points must be considered and some questions answered.

Abstract

L.18. in nine trials performed three times on each farm.

L.28-31. Improve writing.

Introduction

L.37. Change "prices" to "figures".

L.49. Change "Accordingly" to "Consequently".

L60-61. It is important to include a sentence that allows this paragraph to be linked to the next one since it talks directly about vaccines without a background that justifies their use.

Methods 

L.104. Where were they shipped? Be more specific.

L.107-110. Please reword as the sentence is very ambiguous.

L.110. Change "contains" to "contained".

L.112. Why 9 or 10?

L.116. Table 1: To not be so repetitive, authors can combine table rows.

L.126. A fecal sample of 1 gram was homogenized...

L.147-148. Check font format. 

L.177. The lesion score is not a quantitative variable but rather an ordinal variable. Therefore, a non-parametric analysis must be performed. Please make the corresponding changes to the analysis.

Results

L.194. Table 2: Review the comment on the type of analysis that should be performed (Non-parametric). Furthermore, considering this behavior where the standard deviation is very high, it is not completed with the assumptions of homoscedasticity (homogeneity of variance) and normality, so a parametric analysis could not be carried out. Place the results as mean +- standard error.

The number of oocysts will have to be separated into an independent table since this can be analyzed by one-way ANOVA.  Are the results shown on day 1?

L.199. Figure 2 is very difficult to understand, perhaps the use of tables is a better option. The text describing the results establishes lesion scores considering different days and that is not comparable, please restructure this section. Perform the corresponding statistical analysis including significance.

L.201. Please present the result as the mean +- standard error.

L.223. Do these average values ​​come from the different days of analysis or per day and considering the 3 independent studies?

If the entire analysis is considered, it is logical that there are very high standard deviations, but it is not representative and results would be overestimated.

L.240. Table 3: Are the assumptions of normality and homogeneity of variances met to be able to perform parametric analysis? the standard deviation is very high.

Discussion 

L.337. Improve the way in which the results are discussed and contrasted since it is written as if the authors had carried out the study with which they contrast their results.

L.368-371. Although this is correct, how would the parameters evaluated during an infection process be affected? Is the vaccine protecting? what is the amount of oocysts in the vaccine?

L.381. And in a process of infection with coccidia would it protect the host? At what age are birds most susceptible to being infected by coccidia?

L.400. Change "region" to "section".

L.402-403. Were there significant differences in the results or not?

L.426. In the present study, 40 weeks were not reached, this information could not be used to compare the results.

Author Response

We sincerely appreciate the editor and reviewers for their valuable time, and insightful recommendations for enhancing our manuscript. The authors have diligently considered the comments, and below are our point-for-point responses. We trust these will align with your expectations and contribute to the further improvement of our manuscript.

Reviewer’s comments 3: 

While the article is interesting, there are errors in the results analysis section that need to be resolved. In addition, some specific points must be considered and some questions answered.

Abstract

1. L.18. in nine trials performed three times on each farm.

Response: The indicated error was revised.

2. L.28-31. Improve writing.

Response: The indicated sentence was modified as “This large-scale field trial of a live Eimeria vaccine indicates that coccidiosis vaccines can have meaningful effects on the poultry industry and could be used as an alternative to prophylactic use of anticoccidial drugs under field conditions” in line 28.

Introduction

3. L.37. Change "prices" to "figures".

Response: The indicated error was corrected.

4.L.49. Change "Accordingly" to "Consequently".

Response: The indicated error was corrected.

5. L60-61. It is important to include a sentence that allows this paragraph to be linked to the next one since it talks directly about vaccines without a background that justifies their use.

Response: The additional sentence was added as “Of them, vaccines that do not cause any resistance are receiving great attention in the chicken industry to prevent coccidiosis” in line 62.

Methods 

6. L.104. Where were they shipped? Be more specific.

Response: “they were shipped” was changed with “they were shipped to slaughterhouse” in line 106.

7. L.107-110. Please reword as the sentence is very ambiguous.

Response: The indicated sentences were rephrased as “The commercial Fortegra coccidiosis vaccine was provided free of charge by MSD Animal Health Korea Ltd. Coccidiosis vaccine was sprayed coarsely on 1-day-old chicks at the hatchery using a suitable device according to the manufacturer’s instructions. Vaccinated chicks were transported to the trial farms in approximately two hours by a special vehicle” in line 109.

8. L.110. Change "contains" to "contained".

Response: The indicated error was corrected.

9. L.112. Why 9 or 10?

Response: One of the three trial farms has three buildings. Three samples were collected from each building, resulting in a total of nine samples. A total of 10 samples were collected from the other two farms. Therefore, the number of samples was n = 9 to 10.

10. L.116. Table 1: To not be so repetitive, authors can combine table rows.

Response: Table 1 was modified according to suggestion.

11.L.126. A fecal sample of 1 gram was homogenized...

Response: The indicated error was corrected.

12. L.147-148. Check font format. 

Response: The font format was corrected in line 150.

13. L.177. The lesion score is not a quantitative variable but rather an ordinal variable. Therefore, a non-parametric analysis must be performed. Please make the corresponding changes to the analysis.

Response: Table 2 for lesion score was modified with data obtained from a non-parametric analysis and presented as mean ± standard error. The corresponding changes were carried out in line 194.

Results

14. L.194. Table 2: Review the comment on the type of analysis that should be performed (Non-parametric). Furthermore, considering this behavior where the standard deviation is very high, it is not completed with the assumptions of homoscedasticity (homogeneity of variance) and normality, so a parametric analysis could not be carried out. Place the results as mean +- standard error. The number of oocysts will have to be separated into an independent table since this can be analyzed by one-way ANOVA. Are the results shown on day 1?

Response: We appreciate the reviewer's comment. Table 2 separated into Table 2 for lesion scores and Table 3 for fecal oocyst shedding. Table 2 for lesion score was modified with data obtained from a non-parametric analysis and presented as mean ± standard error. The corresponding changes were carried out in lines 193~196 and 204~216. Oocysts can be identified in the feces approximately 5 to 6 days after inoculation with E. acervulina. In case of inoculation with E. maxima and E. tenella, oocysts can be identified in the feces approximately 6 to 7 days. Therefore, 1-day-old samples collected after vaccination have no useful information regarding oocyst and lesion scores.

15. L.199. Figure 2 is very difficult to understand, perhaps the use of tables is a better option. The text describing the results establishes lesion scores considering different days and that is not comparable, please restructure this section. Perform the corresponding statistical analysis including significance.

Response: The supplementary table 1 (Table S1) containing related data has been added to improve understanding of Figure 2. According to the suggestion, Figure 2 was also modified to mean ± standard error. The text describing lesion scores was corrected. When broiler chickens are raised on farms where coccidia occurs without using anti-coccidial agents, production performance is very poor and chickens are generally vulnerable to secondary infections. As a result, farm productivity is so low that it is difficult to realistically create a control group. The oocysts present in coccidia vaccines vary greatly in amplification speed and amount depending on the farm conditions. Therefore, the lesion score values ​​were compared and analyzed as shown in Table 2.

16. L.201. Please present the result as the mean +- standard error.

Response: The indicated values were changed with mean ± standard error in lines 204~216.

17. L.223. Do these average values ​​come from the different days of analysis or per day and considering the 3 independent studies? If the entire analysis is considered, it is logical that there are very high standard deviations, but it is not representative and results would be overestimated.

Response: Average values ​​for Table 2 and 3 were obtained using results of different dates of 3 independent trials. Thus, Tables 2 and 3, which comprehensively analyze data at all time points after vaccination, are supported by Figures 2 and 3, which display data at each time point.

Unlike to virus or bacterial vaccines, live coccidiosis vaccines do not infect all chickens at once. Therefore, there is a process in which oocysts released from an infected chicken induce immunity by re-infecting or inducing infection in other chickens. This immunization method takes approximately 3 weeks. Since the effectiveness of coccidia vaccines is affected by various conditions such as chicken breeding density, humidity and temperature, it is believed that comparing the results as a whole rather than comparing the results of a specific period can reduce mistakes, especially in farm trials.

18. L.240. Table 4: Are the assumptions of normality and homogeneity of variances met to be able to perform parametric analysis?

Response: To reduce variation, the experiment was conducted on farms run by the same company, and recent data from the same farms was used for comparison. Since the two groups were compared, t-test was used. The production index (PI) used in Table 4 had a very high standard deviation. As the exact cause is currently unknown, it is judged to need to be considered in future research.

Discussion 

19. L.337. Improve the way in which the results are discussed and contrasted since it is written as if the authors had carried out the study with which they contrast their results.

Response: The section of discussion was improved according to reviewer’s suggestion.

20. L.368-371. Although this is correct, how would the parameters evaluated during an infection process be affected? Is the vaccine protecting? what is the many of oocysts in the vaccine?

Response: If a problem occurs with the vaccine, the intestinal lesion score increases, body weight decreases, and explosive oocyst production occurs. Vaccine can protect chickens. Without vaccination or anti-coccidiosis drugs, it is very difficult to raise broiler chickens on farms. The vaccine used in these trials contains 1,500 oocysts containing four different Eimeria species. It contains 600 oocysts from E. acervulina, 300 oocysts from E. maxima, 400 oocysts from E. mivati, and 200 oocysts from E. tenella, respectively.

21. L.381. And in a process of infection with coccidia would it protect the host? At what age are birds most susceptible to being infected by coccidia?

Response: If chickens are infected with very large numbers of oocysts, especially highly virulent strains such as E. tenella, the host is at risk. Therefore, chickens for vaccination are infected with a very small number of sporulated oocysts through spray. Chickens of any age without immunity to coccidia are susceptible regardless of age.

 22. L.400. Change "region" to "section".

Response: The indication was changed.

23. L.402-403. Were there significant differences in the results or not?

Response: As shown in Table 2, there were no significant differences in all results except for JH farm in the duodenum section. In addition, the intestinal lesion score for each chicken was assigned a numerical value from 0 to 4. It is very difficult to detect pathological symptoms at a lesion score of less than 1, and it is generally close to normal.

24. L.426. In the present study, 40 weeks were not reached, this information could not be used to compare the results.

Response: The indicated sentence and reference were removed.

Round 2

Reviewer 3 Report

Comments and Suggestions for Authors

Table 1. Capital letter "Open...."

L.184. Are the authors sure that it is Dunnett's test and not Dunn's?

Author Response

Manuscript ID: Vaccines-3090856R2

We very much appreciate the constructive comments and suggestions of the reviewers, and are grateful, for the thorough examination of our data and recommendations for improvement. Below are our point-for-point responses. We hope these will meet with your approval.

Comments and Suggestions:

Q1. Table 1. Capital letter "Open...."

Response: The indicated word was corrected with capital letter in Table 1.

Q2. L.184. Are the authors sure that it is Dunnett's test and not Dunn's?

Response: The indicated error was corrected with “Dunn's” in line 189.
